# A Study of State Aliasing in Structured Prediction with RNNs

**Layla El Asri, Adam Trischler**
Microsoft Research Montreal
Montreal, Canada
`{first.last}@microsoft.com`

## Abstract

End-to-end reinforcement learning agents learn a state representation and a policy at the same time. Recurrent neural networks (RNNs) have been trained successfully as reinforcement learning agents in settings like dialogue that require structured prediction. In this paper, we investigate the representations learned by RNN-based agents when trained with both policy gradient and value-based methods. We show through extensive experiments and analysis that, when trained with policy gradient, recurrent neural networks often fail to learn a state representation that leads to an optimal policy in settings where the same action should be taken at different states. To explain this failure, we highlight the problem of state aliasing, which entails conflating two or more distinct states in the representation space. We demonstrate that state aliasing occurs when several states share the same optimal action and the agent is trained via policy gradient. We characterize this phenomenon through experiments on a simple maze setting and a more complex text-based game, and make recommendations for training RNNs with reinforcement learning.

## 1 Introduction

RNNs have been successfully trained with RL for structured prediction tasks like dialogue and Natural Language Generation (NLG) (Ranzato et al., 2016; Bahdanau et al., 2017; Strub et al., 2017; Narayan et al., 2018; Wu et al., 2018). It has become fairly common practice first to train these models on data with the maximum-likelihood objective and then to continue their training as RL agents with additional objectives (e.g., maximizing the BLEU score; Papineni et al. 2002). Most commonly, this RL training is based on policy-gradient methods. Several motivations for this setup are often put forward, including the following: (1) teacher forcing for maximum likelihood induces an *exposure bias*, since at each time step of decoding, the model is trained to output the most likely prediction given previous outputs from the *reference* sequence, not the outputs predicted by the model itself; (2) policy-gradient methods like REINFORCE (Williams, 1992) enable optimizing non-differentiable objectives, an especially useful property in discrete structured-prediction settings like language.

In the NLG and dialogue settings, an agent's actions consist of sequences of tokens (e.g., words or characters). When the model is trained with RL, typically, the agent follows one of two strategies. The first is to output a sequence of tokens via beam search and thus greedily follow the policy learned with the maximum likelihood objective. The second is to sample a token from the output distribution at each time step, which allows for some exploration of the action space. Recent work has suggested that sampling yields better results (Strub et al., 2017; Wu et al., 2018), since exploration might help the model to correct bad behaviour learned during max-likelihood training. In this paper, we describe a distinct problem specific to RNNs that exploration also helps to mitigate: state aliasing.

State aliasing occurs when two or more distinct states are conflated in a model's representation space (McCallum, 1996). In certain environments, this problem impedes the agent in learning the optimal policy. We highlight the problem of state aliasing in RNNs trained with policy gradient-based methods. We study how this problem affects learning in a simple maze with atomic actions, a structured-prediction maze setting where actions are compound, and in a text-based game. On

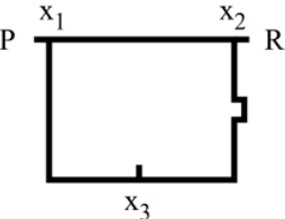

Figure 1: Simple maze. Source: (McCallum, 1996)

|       | right | left  |
|-------|-------|-------|
| $x_1$ | $x_3$ | $x_3$ |
| $x_2$ | $x_3$ | $x_3$ |
| $x_3$ | $x_1$ | $x_2$ |

Table 1: Transition dynamics for the maze.

|       | right | left  |
|-------|-------|-------|
| $x_1$ | 0.7   | -1    |
| $x_2$ | +1    | -1.3  |
| $x_3$ | -0.5  | -0.7  |

Table 2: Reward function for the maze.

the simple maze, we show that RNNs sometimes fail to disambiguate states, and worse, confound states if they share the same optimal action. We design a text-based game as a proxy for dialogue and demonstrate that state aliasing occurs here when the same action is taken several times during the game. Finally, we demonstrate that regularization, through entropy-based exploration or value-function estimation, helps to alleviate this issue.

## 2 A SIMPLE MAZE ENVIRONMENT

### 2.1 ENVIRONMENT

McCallum (1996) used an example to show that if a state representation aliases states based on their optimal action, it might not be possible to retrieve the optimal policy from the representation. His example is reproduced in Figure 1. It is a simple three-state Markov decision process with states $x_1, x_2$, and $x_3$. The agent always starts an episode in state $x_3$ facing south. From $x_3$, it can either travel to $x_1$ by going right or to $x_2$ by going left. It costs a little more to go to $x_2$ because the path is longer. From $x_1$ or $x_2$, the agent always travels back to $x_3$. The agent can only choose between going left and going right, so it need only make a decision when it faces a T-shaped intersection. The intermediate rewards are a combination of the distance travelled by the agent and a special reward obtained when going through one of the two dead-ends.

The transition dynamics are given in Table 1 and the rewards are described in Table 2. The state transitions are deterministic and the rewards are known. It is thus possible to compute the optimal policy for a given horizon with dynamic programming. The optimal policy is to go left in $x_3$ and then right in $x_2$. Now let us consider an agent that does not have access to the true state but instead can only observe whether it is facing north or south. This state representation aliases $x_1$ and $x_2$ (in both cases, the agent faces north). With this representation, the optimal policy is to go left when facing south and to go right when facing north. However, McCallum shows that computing expected returns when $x_1$ and $x_2$ are aliased leads to learning to go right both when facing south and when facing north, which is suboptimal. We refer to McCallum's thesis for full computations.

A recurring and open problem in dialogue generation with RNNs is repetition (Das et al., 2017; Strub et al., 2017; Li et al., 2016). We hypothesize that this problem might arise from state aliasing in the model: if the model confounds the observation sequences for distinct states into the same representation, then the model will output the same sequence of tokens in those distinct states. The simple maze is a good way to test for this hypothesis: we know that if a model trained on this maze confounds the observation sequences for $x_3, x_2$ and $x_3, x_1$, then it will fail to learn the optimal policy. The simplicity of the task enables us to analyze what the model learns and why it fails. We further hypothesize that models trained with policy gradient are susceptible to state aliasing. The reason for this hypothesis is that the output of a model trained with policy gradient is a distribution over actions. This distribution can be very similar for two different states which share the same optimal action (i.e, a probability close to 1 for the optimal action and 0 for the others). Thus, if the policy is the same for two different states, an encoder RNN trained by policy gradients and backpropagation might represent the two states similarly in hidden space. We test this hypothesis in the remainder of the paper.

| Architecture | Algorithm | Exploration Method | Number of Failures |
|---|---|---|---|
| LSTM | REINFORCE | None / Entropy | 14 / 6 |
| LSTM + baseline (not shared) | REINFORCE | None / Entropy | 17 / 17 |
| LSTM + baseline (shared) | REINFORCE | None / Entropy | 11 / 7 |
| **GRU** | **REINFORCE** | None / **Entropy** | 10 / **1** |
| **GRU + baseline (not shared)** | **REINFORCE** | **None / Entropy** | **0 / 0** |
| **GRU + baseline (shared)** | **REINFORCE** | **None / Entropy** | **0  0** |
| **LSTM** | **DQN** | $\epsilon$-**greedy** | **5** |
| **GRU** | **DQN** | $\epsilon$-**greedy** | **2** |
| **Logistic Regression** | **REINFORCE** | **None** | **0** |
| **MLP** | **REINFORCE** | **None** | **0** |

Table 3: Number of times the different models and algorithms fail to learn the optimal policy on the simple maze out of 50 runs. *Not shared* (as in parameters not shared) means that the baseline is computed via another model whereas *shared* means that the baseline is computed by the same model in a multi-task fashion.

## 2.2 MODELS

We train several RNN variants on the simple maze and analyze how they represent the different states. Our base models are a GRU (Gated Recurrent Unit, Cho et al. (2014)) and an LSTM (Hochreiter & Schmidhuber, 1997). We train them with two algorithms: REINFORCE (Williams, 1992) and DQN (Mnih et al., 2013). For REINFORCE, we run experiments with and without a baseline (Williams, 1992). When a baseline is added, we test two settings: one where the baseline is computed by the same model that makes decisions, and one where the baseline is computed by a different model. In both settings, the baseline estimates $E_{\sim\pi}[\sum_t R_t \mid s_t]$, where $s_t$ is the state at time step $t$ and $\sum_t R_t$ is the sum of rewards obtained by the agent after visiting $s_t$ and following policy $\pi$. Complete details of the models and the training algorithms are given in the appendix.

The input to the models is the current state of the agent, represented by a 3-dimensional one-hot vector: $[0, 0, 1]$ for $x_3$, $[1, 0, 0]$ for $x_1$, and $[0, 1, 0]$ for $x_2$. The RNNs map this input to a real-valued 2-dimensional hidden state. A linear transformation is then applied to the hidden state. In the case of DQN, this output represents the values of the two actions *left* or *right*. For REINFORCE, an additional softmax function outputs a probability distribution over the two actions. We run 2000 episodes of 2 steps with 50 different random seeds. In each episode, the agent starts in state $x_3$, makes one decision that leads it to $x_1$ or $x_2$, and then makes one more decision that always leads it back to $x_3$.

## 2.3 RESULTS

A first observation is that in all our experiments, the agents either discovered the optimal policy or converged upon the same suboptimal policy as described by McCallum (1996) when the state representation aliases $x_1$ and $x_2$. Given this observation, instead of reporting the average return obtained by the agent, we report the number of failures to learn the optimal policy over the 50 runs for the different architectures and algorithms. This is shown in Table 3.

An obvious trend is that no matter the setting, the LSTM fails more than the GRU. An explanation – which we validated empirically but do not plot here for brevity – is that the LSTM takes longer to learn. Without any exploration strategy nor baseline, the LSTM and the GRU fail 28% and 20% of the time, respectively. Our hypothesis is that models fail because they learn a representation that aliases $x_1$ and $x_2$, since these two states share the same optimal action. For verification, we run another experiment where we invert the rewards for state $x_1$ such that its optimal action is no longer the same as that for $x_2$. In this setting, the LSTM never failed and the GRU failed only 4 times over 50 runs. We further compute the Euclidean distance between the hidden state computed after visiting $x_3$ and then $x_1$, and the hidden state computed after visiting $x_3$ and then $x_2$: $||h_{x_3,x_1} - h_{x_3,x_2}||^2$. We plot the evolution of this quantity during a run when the agent learns the optimal policy (bottom of Figure 2) and during a run when it converges to the suboptimal policy (top of Figure 1) with

the GRU-based model with no baseline and no exploration strategy. We also plot the probability of taking the optimal action in $x_3$. These two quantities are tightly linked. It is when the distance between $h_{x_3,x_1}$ and $h_{x_3,x_2}$ becomes larger that the probability of taking the optimal action in $x_3$ grows. In the case when the agent learns the suboptimal policy, the distance between the two hidden states quickly goes to 0 and the agent cannot learn the optimal policy because of state aliasing. In the case when it learns the optimal policy, the difference between hidden states grows sufficiently so that the model can distinguish the two states and compute the right policy. An interesting observation is that once it has learned the optimal policy, the model seems to alias $h_{x_3,x_1}$ and $h_{x_3,x_2}$ again since the distance goes down to close to 0. This aliasing is without consequences because once the optimal policy is learned the model does not visit $x_1$ anymore. We leave the explanation of this phenomenon for future work.

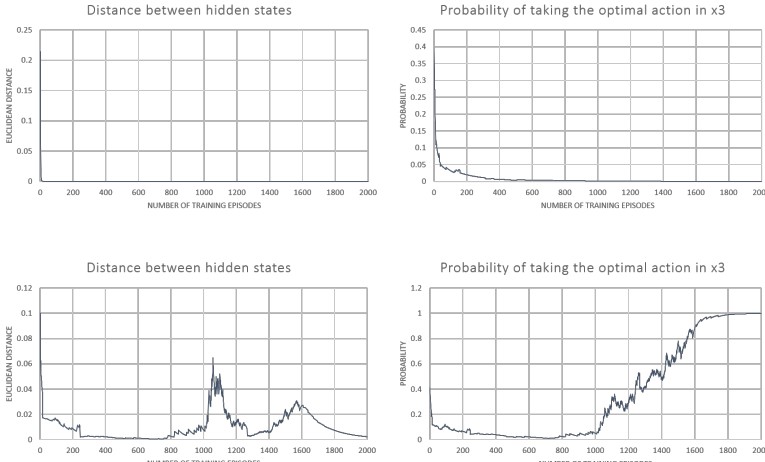

Figure 2: On the left: evolution of the distance between hidden states $h_{x_3,x_1}$ and $h_{x_3,x_2}$ throughout learning. On the right: evolution of the probability that the agent takes the optimal action in state $x_3$ throughout learning. On the top figures, the agent learns a suboptimal policy. On the bottom figures, the agent learns the optimal policy.

Results in Table 3 also suggest that adding exploration or a baseline both help to avoid state aliasing in REINFORCE-based training. In the case of the GRU, the failure rate drops to between 0 and 2% when a baseline and/or entropy-based exploration is added. In the case of the LSTM, entropy-based exploration gives the largest improvement. Note that we also used REINFORCE to train an agent (referred to as Logistic Regression in the table) that maps the input directly to a probability distribution over actions, rather than computing an intermediate representation as in an RNN. This agent never suffers from state aliasing and does not require entropy-based exploration to learn the optimal policy consistently. Similarly, we trained a one-layer MLP to check if state aliasing occurred in other non-linear networks. The MLP model consistently learned the optimal policy without any exploration. This suggests that the problem is specific to RNNs.

The second part of our hypothesis is that value-based RL methods, like DQN, are less susceptible to the aliasing problem. Note that, given the simplicity of the task, we do not use experience replay in any of our experiments with DQN. As shown in Table 3, the GRU and the LSTM only fail 2% and 4% of the time respectively when trained with DQN and an exponentially-decaying $\epsilon$-greedy policy. In our experiments, we monitored the Euclidean distance between the hidden states and observed that the failure cases are not due to state aliasing but to insufficient exploration within the 2000 training episodes. DQN avoids state aliasing because it computes a representation of the states that is based on the action values instead of the probability of taking the actions. Since the rewards are different for $x_1$ and $x_2$, it is not possible to alias these states and at the same time compute correctly the values of the state-action pairs.

From these experiments, we draw the following conclusions: (1) the GRU and the LSTM are both susceptible to state aliasing when they learn a representation of states based on policy gradients;

(2) regularization, via a baseline or entropy-based exploration, helps to avoid the aliasing problem. These experiments highlight the fact that RNNs alias states in fully observable environments. The state is fully observable and yet $x_1$ and $x_2$ are aliased in the RNNs' internal representation. Since the RNNs learn the optimal policy significantly more consistently when trained with a value-based method (DQN) or when $x_1$ and $x_2$ do not share the optimal policy, we can confirm that aliasing occurs with policy-gradient training when states share the same optimal action. In the next section, we complexify the action space to a structured-prediction setting.

## 3 PREDICTING SEQUENCES OF TOKENS

We extend the previous setting to a structured prediction problem with sequential actions. We modify the maze so that the agent must output a sequence of two tokens to move on to the next state. The transition dynamics are given in Table 5. As in the previous maze, we run two-step episodes starting from state $x_3$. We define three reward functions to test for state aliasing. These functions are described in Table 4.

| Function $R_1$ | r,r | r,l | l,l | l,r |
|---|---|---|---|---|
| $x_1$ | -1 | -1 | 0.7 | 0.7 |
| $x_2$ | 1 | 1 | -1.3 | -1.3 |
| $x_3$ | -0.5 | -0.5 | -0.7 | -0.7 |

| Function $R_2$ | | | | |
|---|---|---|---|---|
| $x_1$ | 0.7 | 0.7 | -1 | -1 |
| $x_2$ | 1 | 1 | -1.3 | -1.3 |
| $x_3$ | -0.5 | -0.5 | -0.7 | -0.7 |

| Function $R_3$ | | | | |
|---|---|---|---|---|
| $x_1$ | 0.7 | -1 | -1 | -1 |
| $x_2$ | -1.3 | 1 | -1.3 | -1.3 |
| $x_3$ | -0.5 | -0.5 | -0.7 | -0.7 |

| state \ action | r,r | r,l | l,l | l,r |
|---|---|---|---|---|
| $x_1$ | $x_3$ | $x_3$ | $x_3$ | $x_3$ |
| $x_2$ | $x_3$ | $x_3$ | $x_3$ | $x_3$ |
| $x_3$ | $x_1$ | $x_1$ | $x_2$ | $x_2$ |

Table 4: Reward functions for the sequential problem. *r* is short for right and *l* is short for left.

Table 5: Transition dynamics for the sequential problem. *r* is short for right and *l* is short for left.

The optimal policy given reward function $R_1$ is to output a sequence of items starting with *left* in $x_3$, and then output a sequence of items starting with *right* in $x_2$. In this case, the optimal actions in $x_1$ and $x_2$ are distinct: if the agent is in $x_1$, it should output a sequence of tokens starting with *left*. Function $R_2$ entails the same optimal policy but changes the optimal actions in $x_1$ so that they correspond exactly to the optimal actions in $x_2$. Finally, $R_3$ is meant to test if state aliasing occurs when only the first token of the optimal actions is shared between $x_1$ and $x_2$. In this case, in $x_1$, the agent should output *right, right* whereas in $x_2$, it should output *right, left*. We train a simple sequence-to-sequence model with a GRU encoder that takes the 3-bit input and outputs a 3-dimensional hidden state. This state is used to initialize a GRU decoder, which takes as input the *empty* action (-1) at the initial time step and the previously produced token thereafter (0 for *right*, 1 for *left*). The decoder uses a 3-dimensional hidden state. We apply a linear transformation and then a softmax on this hidden state to output a probability distribution over the next action token. We train this model via REINFORCE without entropy-based exploration nor any baseline computation. Additional implementation details are provided in the Appendix. We report the number of failures over 50 runs in Table 6.

When the optimal actions are distinct (function $R_1$), the encoder-decoder model only fails 5 times out of 50, which is similar to our observations in the single-token case. When the optimal actions are exactly the same in $x_1$ and $x_2$ (function $R_2$), the model almost always fails to learn the optimal policy (92% of the time). In our experiments, whenever the model fails, it converges to the same suboptimal policy due to state aliasing – the hidden states of the encoder after visiting $x_1$ and after visiting $x_2$ are very close. Surprisingly, the model is subject to state aliasing even when only the first token of the optimal action is the same for $x_1$ and $x_2$ (function $R_3$). The problem does not

| Reward Function | Number of failures |
|:---:|:---:|
| $R_1$ | 5 |
| $R_2$ | 46 |
| $R_3$ | 15 |

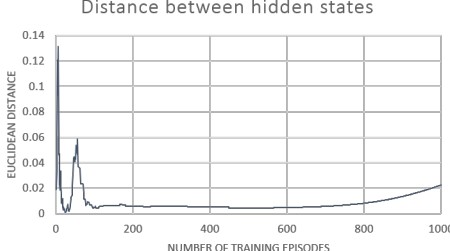

Distance between hidden states

Table 6: Number of times the model fails to learn the optimal policy on the sequential maze.

Figure 3: Evolution of the distance between hidden states $h_{x_3,x_1}$ and $h_{x_3,x_2}$ throughout learning with $R_3$

occur as often in this setting, but the model fails 30% of the time. We again plot the evolution of the Euclidean distance $||h_{x_3,x_1} - h_{x_3,x_2}||^2$ during an unsuccessful run with $R_3$, in Figure 3, where $h$ is the hidden state of the encoder. The aliasing phenomenon appears clearly here and our analysis suggests it is responsible for a large number of failure cases when using $R_3$.

This result is suggestive because in dialogue tasks, not only does the same text sequence occur across different dialogues, but many sequences across different turns also share the same initial token (e.g., *I*, *it*, *the*, etc.). Jiang & de Rijke (2018) observed that attention-less encoder-decoder models based on RNNs quickly develop *over-confidence* on the first token to generate. The result is that beam search tends to produce sentences that all start with the same token. Our experiments suggest that this could be linked to state aliasing in the encoder. To support this observation in a natural language setting, we run a third set of experiments on a text-based game.

## 4 EXPERIMENTS ON A TEXT-BASED GAME

We used the maze setting as a minimal example to demonstrate the problem of state aliasing in RNNs. Now we investigate the phenomenon under more complex conditions, using a proxy for human-machine dialogue.

In particular, we study RNN-based agents as they learn to solve text-based games. Text-based games are complex, interactive simulations in which text describes the game state and players make progress by issuing text commands. These games use natural language to describe the state of the world, to accept actions from the player, and to report subsequent changes in the environment. In this way, playing a text-based game can be considered a kind of dialogue with the environment (Côté et al., 2018) – the environment acts as a user simulator (El Asri et al., 2016).

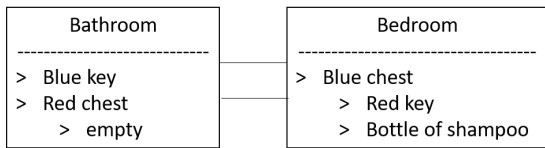

Figure 4: TextWorld game used to illustrate state aliasing in a language setting.

The maze experiments show that state aliasing arises in GRUs and LSTMs when different states share the same optimal action and the networks are trained with policy-gradient methods. To study this phenomenon in a setting like dialogue modelling, we design a game in which the same action must be taken in different contexts. We use the TextWorld framework (Côté et al., 2018) to build the game. The setup is depicted on Figure 4. There are two rooms connected by a hallway. The agent starts in the bedroom and must find a bottle of shampoo to place in a chest in the bathroom. To accomplish this goal, the agent must perform the following sequence of 11 actions: {*go west, take the blue key, go east, unlock the blue chest with the blue key, open the blue chest, take the red key, take the bottle of shampoo, go west, unlock the red chest with the red key, open the red chest, insert the bottle of shampoo into the red chest*}. In executing this sequence, the agent must *go west*

twice: at the very beginning of the game and once it has recovered the contents of the blue chest. We explore whether this causes issues for an RNN-based agent trained through policy gradient.

To focus on the representation, we train a retrieval-based model rather than a generative one. The retrieval model receives a list of valid actions from the TextWorld game engine at every time step (which may change across steps) and learns to select the action that moves it towards the objective. Each valid action is a sequence of tokens like {*open the blue chest*}. The model induces a probability distribution over the action list at each time step, which it samples from greedily or otherwise.

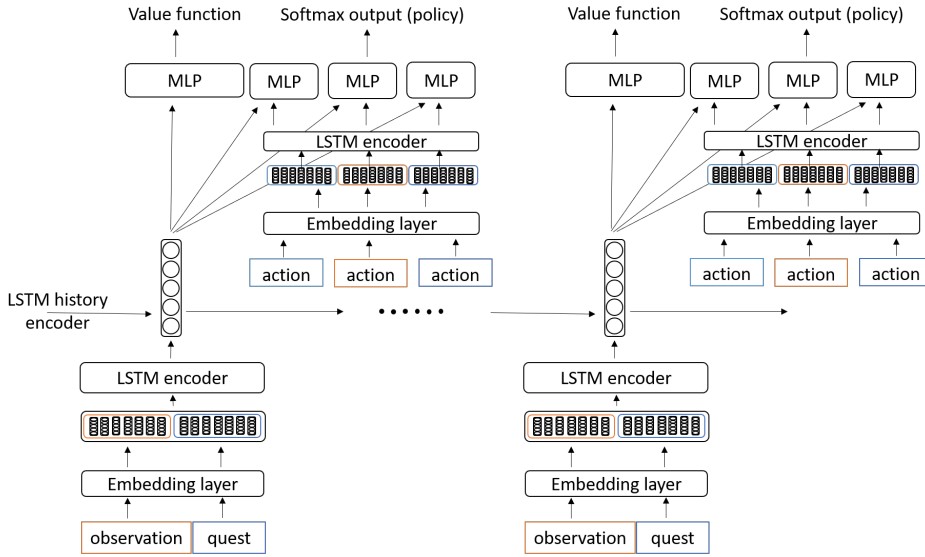

Figure 5: Retrieval model tested on TextWorld.

The model, inspired by the one proposed by He et al. (2016), is described in Figure 5. At each time step, the game engine returns an observation of the current state of the game. This observation describes the room where the agent is located, the various objects in this room, as well as the objects in the agent's inventory. We concatenate all the sentences in this description, tokenize them into words, map the words through an embedding matrix, and encode the sequence of embeddings via LSTM (the LSTM encoder). We similarly encode the quest that the agent must perform using a separate LSTM encoder with distinct parameters. In our experiments, the quest consists of a short text string that describes the objective, in the form of all the actions the agent must take to complete the game. The concatenation of the quest encoding and the observation encoding is passed to a higher-level LSTM, which encodes game history. We take the hidden state of this LSTM at a given time step as the representation of the game history up to that time step.

The TextWorld game engine returns all valid actions given the current state of the game.[1] We encode each action with an LSTM, similarly to the observations. We then replicate the history encoding and concatenate it with each action encoding, passing these concatenated vectors separately to an MLP.

We pass the MLP outputs for each action as logits to a softmax, which yields a probability distribution $\pi_\theta(s_t, a)$ over actions. The history encoding is also used to predict the value $\hat{V}_\theta(s_t)$ of the current state $s_t$ to use as a baseline for REINFORCE. This is done through another MLP. Further details and parameter values for this model are given in the Appendix.

Each episode starts with the agent in the bedroom with an empty inventory and terminates after 11 steps. We reward the agent with score values from the game engine. The score goes up by 1 whenever the agent takes an action that moves it closer to the goal, and down by 1 when the agent takes an action that moves it farther away. When the score does not increment (the agent takes a neutral action, such as analyzing an object), we assign -1. The value estimation, entropy, and policy

---

[1]Note that the number of valid actions could change from state to state. In the computation, we fold commands into the batch dimension to handle this variability.

objectives, respectively, are computed over each training episode as follows:

$$
\mathcal{L}_v = \frac{1}{n_s} \sum_t \left( || \sum_{t' \geq t} R_{t'} - \hat{V}_\theta(s_t)||^2 \right)
$$
$$
\mathcal{L}_e = \frac{1}{n_s} \sum_t \left( - \sum_a \pi_\theta(s_t, a) \log \pi_\theta(s_t, a) \right) \tag{1}
$$
$$
\mathcal{L}_p = \frac{1}{n_s} \sum_{s_t, a_t} \left( - \log \pi_\theta(s_t, a_t) \left( \sum_{t' \geq t} R_{t'} - \hat{V}_\theta(s_t) \right) \right),
$$

where $n_s$ is the number of steps in the episode. We weight and combine these losses as follows to arrive at our overall training objective:

$$
\mathcal{L} = \mathcal{L}_p + \lambda_v \mathcal{L}_v - \lambda_e \mathcal{L}_e. \tag{2}
$$

We train the agent with three different loss-weight settings and report results in Table 7. We ran 50 distinct runs of 500 training episodes.

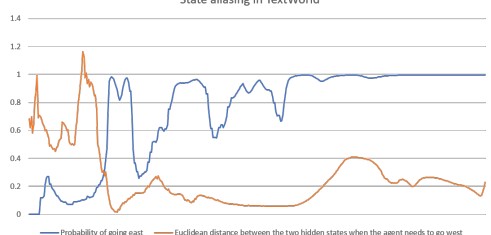

| $\lambda_e$ | $\lambda_v$ | Mean Score (Number of failures) |
|---|---|---|
| 0.5 | 1 | 10.16 (3) |
| 0 | 1 | 7.13 (45) |
| 0.5 | 0 | 9.96 (22) |

Table 7: Results on the TextWorld game.

Figure 6: Illustration of the state aliasing problem in TextWorld.

These experiments confirm that the best performance results when using both entropy-based exploration and a baseline function with shared parameters. In this case as well, exploration seems to bring the greatest improvement over the basic REINFORCE algorithm. We observe interesting state aliasing phenomena. In Figure 6, we plot the Euclidean distance between the hidden representations computed for the states when the agent should go west (twice during an episode), as well as the probability of going east at the last step of the episode (when the agent should insert the bottle of shampoo into the chest). We can see that these two quantities are coupled: when the Euclidean distance starts to decrease, the probability of going east goes up. This is explained by the fact that after the first time the agent must go west, it must take a key and then go east. When it goes west again, there is no key to take but the option of going east exists. Interestingly, the agent learns to unlock the red chest and open it after going west for the second time because this sequence of actions happens after going west in both cases. However, the last action of inserting the shampoo into the chest is distinct, and the agent's policy rapidly peaks towards an action that it has seen previously after going west – in this case, going east. Because of the state aliasing, the probability of going east is high; substantial exploration is needed to learn that going east is not optimal and that the two hidden states should be distinct.

## 5 CONCLUSION

This paper explored the training of RNNs with RL and identified the problem of state aliasing in learning with policy-gradient methods. We propose this analysis as a preliminary step towards better understanding the learning behaviour of RNNs in structured prediction tasks. Based on our findings, we recommend investigating efficient exploration and value-based methods (rather than the currently standard policy gradients) for training RNN-based models for dialogue, language generation, and related problems.

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

APPENDIX

The models trained on the mazes were implemented with PyTorch 0.3.1 and the TextWorld model was implemented with PyTorch 1.0.1. All experiments were computed on a single GPU.

MODELS TRAINED ON THE SIMPLE MAZE

In all cases, if the training algorithm is REINFORCE, a softmax operator is applied to the output of the last linear transformation. We train all models with RMSProp (Tieleman & Hinton, 2012) with a learning rate of 0.01 and we clip the gradient norm to 1.

**RNN Models**    The 3-dimensional one-hot input is passed into one LSTM (resp. GRU) layer. The hidden state of the LSTM (resp. GRU) is used as a representation of the environment state. The hidden state is of dimension 2 and passes through a linear transformation that outputs another 2-dimensional vector. When the baseline shares parameters with the base model, we add a second linear transformation on top of the LSTM (resp. GRU), in parallel. This transformation outputs a scalar. When the baseline does not share parameters with the base model, we train another model that is structurally identical to the base model except that it outputs a scalar instead of a 2-dimensional vector.

**Logistic Regression Model**    This model is a simple linear transformation to transform the 3-dimensional one-hot input to a 2-dimensional probability vector.

**MLP**    This model consists of a linear layer that transforms the 3-dimensional input into 2 dimensions, a $\tanh$ activation, and another linear layer that outputs a 2-dimensional vector.

TRAINING OBJECTIVES AND EXPLORATION STRATEGIES

**REINFORCE**    The agent's goal is to maximize $E_{\sim \pi}[\sum_{t' \geq t} R_{t'} | s_t]$ at each state $s_t$. The corresponding loss is:

$$\mathcal{L}_p = \frac{1}{n_s} \sum_{s_t, a_t} \left( - \log \pi_\theta(s_t, a_t) \left( \sum_{t' \geq t} R_{t'} - \hat{V}_\theta(s_t) \right) \right),$$

where $n_s = 2$ is the number of steps, $\pi_\theta(s_t, a_t)$ is the output of the softmax operator for the action $a_t$, and $\hat{V}_\theta(s_t)$ is the baseline function. If the model is trained without a baseline function, the loss is:

$$\mathcal{L}_p = \frac{1}{n_s} \sum_{s_t, a_t} \left( - \log \pi_\theta(s_t, a_t) \left( \sum_{t' \geq t} R_{t'} \right) \right).$$

If entropy-based exploration is used, the agent learns to minimize:

$$\mathcal{L}_p - 0.1 \frac{1}{n_s} \sum_{s_t} \left( - \sum_a \pi_\theta(s_t, a) \log \pi_\theta(s_t, a) \right).$$

During training, we sample actions from the softmax output; during testing, we choose the action with the highest probability.

**DQN** The agent learns to minimize the following loss:

$$\mathcal{L}_q = f\left(\left[\hat{Q}(s_t, a_t) - (R_t + max_a\hat{Q}(s_{t+1}, a))\right]_{s_t, a_t, s_{t+1}}\right),$$

where $f$ is PyTorch's smoothed L1 norm and $\hat{Q}(s_t, a_t)$ is the output of the last linear transformation for the action $a_t$.

When $\epsilon$-greedy exploration is used, with probability $\epsilon$, the agent performs the optimal action according to its current policy; with probability $1 - \epsilon$, it chooses an action uniformly at random. In all our experiments, $\epsilon$ evolved as follows:

$$\epsilon = \epsilon_{\text{end}} + (\epsilon_{\text{start}} - \epsilon_{\text{end}})\exp\frac{-n_e}{\epsilon_{\text{decay}}}, \tag{3}$$

where $n_e$ is the number of training episodes in the current run, $\epsilon_{\text{end}} = 0$, $\epsilon_{\text{start}} = 0.4$, and $\epsilon_{\text{decay}} = 500$.

MODEL TRAINED ON THE SEQUENTIAL MAZE

The model is an encoder-decoder network. The 3-dimensional input goes through a GRU which outputs a 3-dimensional hidden state. This hidden state initializes the hidden state of a GRU decoder which takes as input the *empty* action (-1). This GRU decoder also uses a 3-dimensional hidden state, which is passed through a linear function returning a 2-dimensional logit vector. This vector is passed through a softmax operator. An action is sampled based on this distribution and then given as input to the decoder at the next time step. We trained this model with REINFORCE without a baseline function nor entropy-based exploration. We use RMSProp with a learning rate of 0.01 and we clip the gradient norm to 1.

TEXTWORLD MODEL

The model is depicted on Figure 5. Word embeddings were initialized according to a uniform distribution between -0.05 and 0.05. The embedding layer outputs vectors of size 256 for each word. The LSTM encoder, the LSTM history encoder, and the MLP all use hidden states of size 128. The LSTM encoder and the LSTM history encoder both have one layer of LSTM units. The MLP that feeds the softmax operator uses $\tanh$ activation in its single hidden layer. The MLP that estimates the state values is similar except it uses ReLU activation.

We use Adam (Kingma & Ba, 2014) with a learning rate of 0.002 and we clip the gradient norm to 5.

