# OpenReview forum: "A Study of State Aliasing in Structured Prediction with RNNs"
_ICLR.cc/2019/Workshop/drlStructPred — drlStructPred 2019_

### Official Review · AnonReviewer1 · 2019-03-31
**Great paper**

**Rating:** 4
**Confidence:** 2

**Review:**

This paper investigates the representations learned by RNN-based RL agents for solving structured prediction problems when trained with both policy gradient and value-based methods. It studies the conditions leading to state aliasing and highlights strategies to prevent this situation. The hypothesis is that state aliasing happens when different states share the same optimal action and can result into a failure to converge to the optimal policy. The authors study this phenomenon using LSTM and GRU networks on synthetic (toy) environments, and validate their hypothesis.

The paper is clear, well written, and very relevant to the workshop.

---

### Official Review · AnonReviewer4 · 2019-04-05
**Decent study, more experimental analysis needed**

**Rating:** 3
**Confidence:** 2

**Review:**

The paper considers reinforcement learning settings where the states are partially observable and an RNN is used as a reinforcement learning agent.  The paper studies the learned representations for the agent’s policy. It shows that when trained with policy gradient methods, the learned policy could be suboptimal when encountering aliased states sharing the same optimal action.  To side step this problem, the following is suggested:

1) Using value-based learning method such as DQN.
2) Regularization through entropy based exploration and using a baseline for the REINFORCE algorithm.

The hypothesis is verified empirically on three simple, yet illustrative reinforcement learning environments: a simple maze, a structured prediction maze setting, and a text based game simulating a dialog setting.

Strengths:
========

The paper provides more insight into some of the failure cases for training RNN reinforcement learning agents. The paper shows that when using policy gradient methods, the agent may fail to learn the optimal policy perhaps due to state aliasing effect.


Weakness:
=========

1) Many experimental comparisons are missing:

a) DQN experiments are missing for the structured prediction task and the text-based  game setting;
b) Results with entropy based regularization were not reported for the structured prediction task.
c) What is the effect of different loss-weight settings on the results? Why are the failures in the first row of Table 7 fewer than the other two weight settings?

2) The analysis considers a very controlled and simplified setting, but ignores several factors that might affect the hypothesis:

a) Does the choice of the optimizer affect the number of failures?
b) In sequence to sequence tasks, does pre-training with MLE help in learning better state representations?


Clarity:
======

The paper is mostly clear and well written. Some of the figure captions are short and non informative (e.g. table 7). The description for the text-based game starting from page 7 is not clear.  For example:

“The retrieval model receives a list of valid actions at every time step “ -> where does this list come from?

This is only mentioned two paragraphs later: “The Text World game engine returns all valid actions given the current state of the game ”

---

### Official Review · AnonReviewer5 · 2019-04-06
**Questions regarding the simple maze**

**Rating:** 1
**Confidence:** 2

**Review:**

The authors investigate state aliasing of the learned representations of RNNs trained with policy gradient methods. They start with a simple maze example and claim to show that state aliasing occurs when several states share the same optimal action and the agent is trained via policy gradient. They find that this does not occur when the agent is trained with Q learning. Then they extend this to a more complex text based game and find that entropy bonuses and baselines improve performance.

I had great difficulty understanding the conclusions stemming from the simple maze example, which is the core of the paper. McCallum introduces the example to demonstrate that when x1 & x2 are aliased, the optimal policy can be represented, but that learning when specifically using Q learning on the aliased states does not converge to the optimal policy. The issue stems from the fact that Q learning backs up the value from the aliased state. The authors claim that "policy-gradient methods directly estimate a policy instead of estimating a state-action value, [so] we expect models trained with these methods to be sensitive to state aliasing." This is in opposition to the conclusions from (McCallum 1996).

It is my understanding that policy gradient methods would not suffer from the same issue.  Are the authors claiming that PG methods do? Then they should show that with aliased x1 & x2 in the tabular case, it converges to a suboptimal policy.

Furthermore, in Fig 2 bottom, it appears that the distance between the representations for x1 and x2 converges to 0 and the policy converges to the optimal policy. This makes sense as the optimal policy can alias x1 & x2, but goes against the conclusions of the author.

Finally, entropy bonuses and baselines are known to improve performance. The authors don't clearly explain the connection between them and their hypothesis that RNN state aliasing is a problem.

---

### Official Review · AnonReviewer3 · 2019-04-07
**Good starting point for a discussion of failure modes in RL-RNN training**

**Rating:** 3
**Confidence:** 2

**Review:**

In this well written paper the authors look into a particular failure mode of RNN models trained via RL: the case where the state representation fails to properly encode the history of previous observations because the next correct/optimal output is the same for different histories. Thus the learned model fails to distinguish the states and produces the same, potentially non-optimal policy.

The paper is quite specific as it addresses a problem that occurs only when training RNNs using RL methods such as REINFORCE to learn policies for a problem where states can be easily confused. Nevertheless, because of the prevalence of RL methods and the limited insight into failure modes this is a useful line of investigation.

The authors work through 3 examples with growing complexity: A simple 3-state problem where the agent has to visit 2 states and then gets a reward, another 3 state problem, where the agent has to perform two sub-actions to change the state and thus 4 actions to get a reward, and a text-based game to simulate a dialogue system where the agent has to learn to rank possible actions given a textual representation of the observations.

All these problems should be trivial to learn with even a tiny RNN and indeed the authors show that a supervised training never deteriorates. In contrast, RL often fails to learn the optimal policy. The authors investigave both REINFORCE and DQN, baseline and no baseline, as well as a loss that takes entropy into account. In all cases a number of training runs deteriorate.

While I don't see any fundamental shortcomings of this paper I am a bit uneasy about the interpretation. Could it be that the observed phenomenon stems from a particular property of the error surface, where a few steps in the wrong direction might be enough to fall for some non-optimal local minimum? That being said I'd recommend this work for publication. The workshop should be the right place to discuss possible reasons for the observed outcome.

Some smaller details: Tone down the first paragraph of the introduction a bit and define what you mean by a partially observable problem first.
Mention the number of runs (50) in the caption of Table 3.

---

### Decision · Program_Chairs · 2019-04-09
**Acceptance Decision**

Accept